# Protocol for CARES–HAPIN: an ambidirectional cohort study on exposure to environmental tobacco smoke and risk of early childhood caries

Sneha S Patil [1,2], Naveen Puttaswamy [1], Andres Cardenas,[3] Dana Boyd Barr [4], Santu Ghosh,[5] Kalpana Balakrishnan[1]

For numbered affiliations see end of article.

**Correspondence to**
Dr Naveen Puttaswamy;
naveen@ehe.org.in

## ABSTRACT

**Introduction** Prenatal and postnatal exposure to environmental tobacco smoke (ETS) has been linked with early childhood caries (ECC), but the specific molecular mechanisms and pathways remain largely unknown. The Caries Risk from exposure to Environmental tobacco Smoke (CARES) within the Household Air Pollution Intervention Network (HAPIN) study aims to establish the association between ETS and ECC by employing epidemiological and novel biomarker-based approaches. Here, we outline the overall design and rationale of the project.

**Methods and analysis** We will leverage the infrastructure and data from the HAPIN trial (India) to mount the CARES study. In this ambidirectional cohort study, children (n=735, aged: 3–5 years) will undergo ECC examination by a trained dentist using standard criteria and calibrated methods. Structured questionnaires will be used to gather information on sociodemographic variables, dietary habits, oral hygiene, oral health-related quality of life and current exposure to ETS. We will collect non-invasive or minimally invasive biospecimens (i.e., saliva, buccal cells, dried blood spots and urine) from a subset of HAPIN children (n=120) to assess a battery of biomarkers indicative of exposure to ETS, early biological effect and epigenetic modifications. Both self-reported and objective measures of ETS exposure collected longitudinally during in utero and early postnatal periods will be accessed from the HAPIN database. We will apply current science data techniques to assess the association and interrelationships between ETS, ECC, and multiple biomarkers.

**Ethics and dissemination** Information gathered in this research will be published in peer-reviewed journals and summaries will be shared with the key stakeholders as well as patients and their parents/guardians involved in this study. Sri Ramachandra Institute of Higher Education and Research Ethics Board has approved the study protocol (IEC-NI22/JUL/83/82).

**Trial registration number** NCT02944682.

## STRENGTHS AND LIMITATIONS OF THIS STUDY

⇒ Caries Risk from exposure to Environmental tobacco Smoke–Household Air Pollution Intervention Network (CARES–HAPIN) dataset comprehensively captures the established risk factors for early childhood caries (ECC), enabling us to accurately attribute the observed ECC to environmental tobacco smoke (ETS) exposure.

⇒ Using longitudinal data on ETS exposure (in utero and early childhood years), this research will identify the critical window of exposure to ETS associated with a greater risk of childhood caries development.

⇒ First study to delve into the intricate biological pathways that link ETS exposure to ECC by applying multivariate statistics for in-depth analysis of high-dimensional biomarker data.

⇒ Limitation includes the extent to which findings are generalisable to other populations, since it predominantly comprises of children from a low socioeconomic status group.

## INTRODUCTION

The WHO estimates that approximately one-quarter of human diseases globally are attributable to environmental causes that can be obviated.[1] The first 1000 days of life (conception to 2 years) is a discrete window as it marks the beginning of rapid growth. Any disturbances due to environmental insults during this stage may have enduring impacts on health.[2 3] Environmental factors inclusive of chemical contaminants, diet, lifestyle choices, social capital, stress, climate and biological response to exposures collectively make up the exposome, the environmental complement of the genome.[4 5] The genetic code is largely static and non-modifiable, while the epigenome is modifiable and has garnered attention as a target for preventive strategies. This has led to an appreciation of the need to capture data from multiple omics layers using a data-driven approach to assess how environmental exposures influence individuals' health. Emerging evidence indicates that exposure to environmental toxicants such as environmental tobacco smoke (ETS) and metal(loid)s can modify oral health.[6–10]

Oral diseases affect an estimated 3.5 billion people worldwide, with dental caries standing out as the most prevalent condition.[11] The stark inequalities, unparalleled burden and shared risk factors with other non-communicable diseases underline dental caries as a major public health problem.[11] Understanding the role of environmental pollutants is imperative for disease prevention as oral health is an integral aspect of general health.[11] This concern is particularly relevant for early childhood caries (ECC) as it is largely preventable and five times as common as asthma.[12 13] ECC is a dynamic, multifactorial disease driven by biofilm and sugar, leading to phasic demineralisation and remineralisation of tooth structure.[14] In recent years, social determinants of health have gained prominence as crucial drivers of ECC.[15] Furthermore, it is closely linked to the 17 Sustainable Development Goals and strongly connected to six of the 13 health targets in goal 3—Good Health and Well-being.[16] Globally, more than 600 million children have untreated dental caries in primary teeth.[14] This not only causes pain and infection but the ramifications extend to negatively impact children's growth, neurodevelopment, school performance and quality of life—all culminating in tremendous encumbrance on the family and society.[13 17] Notably, ECC is on the rise in racially and ethnically minoritised groups, as well as those living in low- and middle-income nations. Among Indian children, the prevalence of ECC varies between 42% and 63%.[18] This disparity arises from limited access to dental care services, financial barriers, scarcity of dental professionals and lack of awareness about oral health.[13 17] For decades, researchers have meticulously studied the aetiology of ECC and developed strategies to combat it. Despite this, its profound burden persists unabated. It may be prudent to explore environmental risk factors beyond the traditional factors in our quest to unravel this complex condition.

Mounting evidence evinces fetal and early-life exposure to ETS, even at low levels, increases the susceptibility of dental caries in children and adult life.[6–9] A recent meta-analysis has demonstrated that children residing with a smoker had significantly higher odds (prenatal—OR: 1.57 (1.47–1.67); postnatal—OR: 1.72 (1.45–2.05)) of presenting with dental caries.[7] ETS, an ubiquitous pollutant, also known as 'passive smoke' or 'secondhand smoke' refers to the mixture of smoke emitted from a burning tobacco product and exhaled by an active smoker.[19] According to the 2014 US Surgeon General's report, there are numerous detrimental health outcomes related to ETS exposure including sudden infant death syndrome, preterm birth, impaired fetal growth, neurocognitive problems, cardiovascular diseases and respiratory illnesses, all of which can persist into adulthood.[20] Recent systematic reviews and reports suggest that there is 'no safe level' of ETS exposure.[19–22] In India, about 23–39% of the population is exposed to ETS, disproportionately affecting children and adolescent non-smokers.[23] The underlying mechanism linking ETS exposure to ECC pathogenesis remains poorly characterised, though

several pathways including impaired salivary gland function, low levels of serum vitamin C and enhanced *Streptococcus mutans* proliferation are hypothesised to be involved.[24–26] Currently, questionnaires are commonly used for assessing prenatal and postnatal ETS exposures; however, they are inadequate for measuring chemical exposures and suffer from misclassification, recall bias and reporting bias. Moreover, assessing exposure cross-sectionally at a given time point only provides a limited understanding of associations and fails to capture the complete range of lifelong exposures along with limited correlational interpretation. However, epigenetics, particularly DNA methylation (DNAm), offers a potential solution as it allows for the retention of environmental perturbations over time. For example, smoking-associated DNAm markers serve as biomarkers of smoke exposure, including the prenatal period.[27] Additionally, DNA adducts originating from tobacco-specific nitrosamines (TSNAs) can also be used as markers of cumulative ETS exposure and early biological effects.

As yet, in low-resourced settings such as India where exposure magnitudes and ECC prevalence are among the highest, evidence for the role of ETS exposure on dental caries in childhood is lacking. Even less is known about the independent and combined impact of ETS exposure during fetal development and/or early childhood on ECC occurrence. In addition, there have been no published studies that have examined variation in DNAm associated with ETS exposure and ECC. To address this gap, we will adopt epidemiological and mechanistic approaches to test the association between prenatal and postnatal ETS exposure and ECC in HAPIN children in India. This research aligns with the National Institute of Dental and Craniofacial Research and the Indian National Oral Health Program's mission to enhance oral health. Unpacking the complexity of biological pathways involved in environmental pollutants and ECC development and the challenges endured by children and their parents will add a major step towards personalised, precision environmental health and precision dental care.

### Study aims

Using an integrated approach that includes ECC evaluation, ETS exposure measurements and early biological markers of effect and susceptibility, we aim to: (1) determine whether in utero, infant and current exposures to ETS (self-reported and measured) are associated with stages of ECC; (2) assess select biomarkers in saliva, buccal cells, dried blood spot (DBS) and urine indicative of exposure to ETS, oxidative stress, inflammation and epigenetic modifications. A conceptual framework underlying this study is shown in figure 1.

## METHODS
### Study setting and participants

Caries Risk from exposure to Environmental tobacco Smoke (CARES) is an ancillary study within the HAPIN

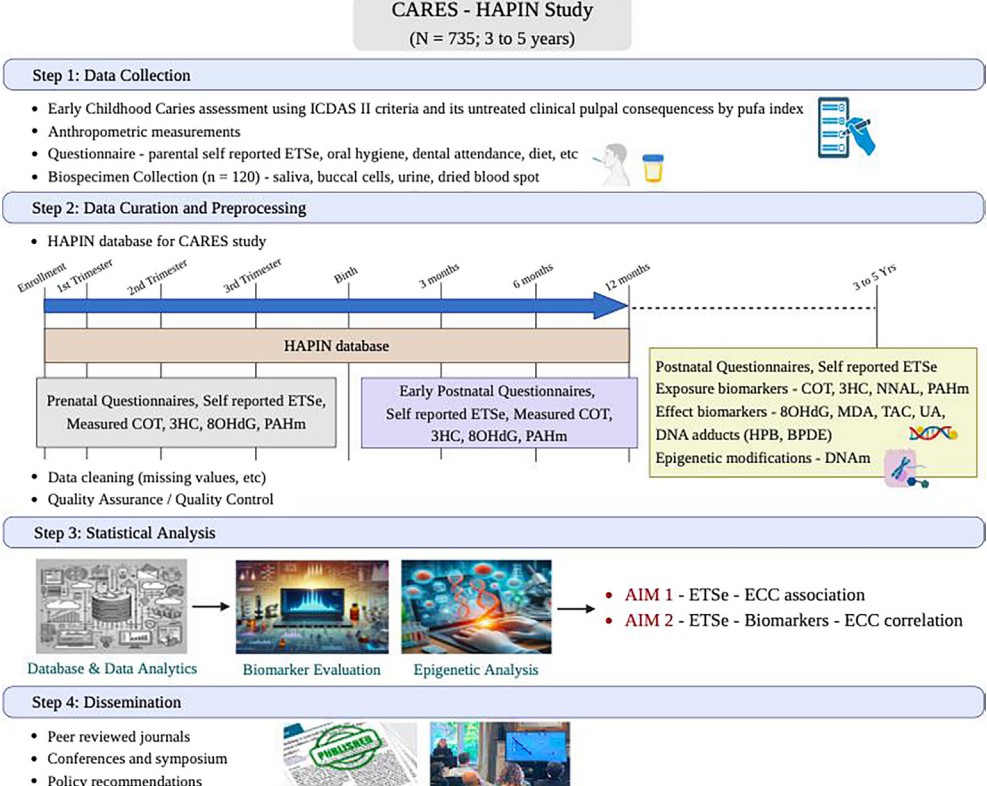

**Figure 1** Conceptual framework of the Caries Risk from exposure to Environmental tobacco Smoke–Household Air Pollution Intervention Network (CARES–HAPIN) study. COT, cotinine; BPDE, benzo[a]pyrene diol epoxide; ECC, early childhood caries; ETS, environmental tobacco smoke; HPB, 4-hydroxy-1-(3-pyridyl)-1-butanone; ICDAS II, International Caries Detection and Assessment System; MDA, malondialdehyde; NNAL, 4-(methylnitrosamino)–1-(3-pyridyl)–1-butanol; PAHm, polycyclic aromatic hydrocarbon metabolite; pufa, pulpal involvement, ulceration, fistula, abscess; TAC, total antioxidant capacity; UA, uric acid; 3HC, 3-hydroxycotinine, 8OHdG, 8-hydroxy-2'-deoxyguanosine

cohort in India. The HAPIN trial (NCT02944682) was launched in 2017 across four diverse countries (India, Rwanda, Guatemala and Peru) to assess the health benefits of clean cooking fuel intervention on birth outcomes, pneumonia and child growth. A detailed description of the HAPIN population and protocol has been previously outlined.[28] In brief, between May 2018 and February 2020, 800 pregnant women were recruited from the Kallakurichi (KK) and Nagapattinam (NP) districts in Tamil Nadu, India. The study enroled women between the ages of 18 and 35 years during antenatal visits at primary health centres in the region, selecting those with a viable singleton pregnancy between 9 and <20 weeks' gestation, who were Tamil-speaking and planning to deliver and receive prenatal care visits at local community clinics or hospitals. Assessments were conducted periodically during enrolment in early pregnancy, 24–28 weeks' gestation, 32–36 weeks' gestation and infancy (3, 6 and 12 months). The study continues to follow-up with mothers and their children, with data collected bi-annually. Of the initial 773 live births at the time of delivery, the HAPIN cohort is currently monitoring 735 children, aged between 3 and 5 years, who will be eligible for inclusion. Kalrayan Hills, KK district, is home to 372 children who live at an altitude of 800 m above sea level, while the blocks of Keezhvelur, Kezhayur, Thalainayiru and Vedaranyam in NP, a coastal district, have 363 children residing at elevations ranging from 10 to 50 m above sea level.

## Data collection
### Clinical dental examination
Dental caries assessment will be undertaken by visual examination using plain mouth mirrors, WHO CPI probes and cotton rolls to remove debris and moisture when required. The diagnosis for each tooth surface will be based on International Caries Detection and Assessment System (ICDAS II) criteria; code 0 indicates no evidence of caries, codes 1–2 initial lesions, codes 3–4 moderate and codes 5–6 extensive caries lesions.[29] If a surface has multiple lesions, it will be coded according to the most severe presentation. To determine individual caries experience, the percentage of affected surfaces out of all examined surfaces will be calculated. The pufa index (presence of pulpal involvement, ulceration, fistula, and abscess) will be employed to evaluate the consequences of untreated dental caries,[30] while the Plaque Index developed by Silness and Loe (1964) will be utilised to assess the plaque levels. The intra-examiner reproducibility will be evaluated through two dental examinations conducted in 30 children at a 15-day interval by a trained and calibrated paediatric dentist. Additional clinical domains will

encompass the evaluation of skeletal and occlusal characteristics, dental trauma and developmental defects of the enamel.

### Anthropometric measurements

Children's height and weight will be measured by the trained surveyors. The US Centers for Disease Control and Prevention growth reference data will be used to compute body mass index (BMI) and Z-scores and assign corresponding BMI categories (such as underweight, normal weight, overweight, and obese).

### ETS exposure assessment

All pregnant women recruited in the HAPIN India site are non-smokers and 32% reported exposure to ETS in the baseline survey. Extant data on measures of cotinine (COT), 3-hydroxycotinine (3HC) and polycyclic aromatic hydrocarbons (PAH) during pregnancy and early childhood (up to 1 year of the child) will be accessed from the HAPIN database. At the time of the ECC assessment, we will capture the smoking status of family members by administering a short questionnaire to the parent(s)/legal guardian(s). Additionally, in a subset of the children, COT, 3HC and PAH metabolites will be measured in saliva and urine to assess current exposure to ETS (details provided in the biomonitoring section). Both self-reported and objective measures of ETS exposure (in utero, infant and current) will be used to assess the risk of caries development.

### Questionnaire

Trained bilingual staff members will interview parents to collect information via detailed questionnaires on oral hygiene (initiation of brushing, frequency, parental supervision, usage of fluoridated toothpaste), diet (frequency of sweet consumption, snacking before sleep), dental attendance (dental visit frequency and reasons), receipt of medications and child's oral health-related quality of life using the Early Childhood Oral Health Impact Scale (ECOHIS). The questionnaire has been designed by the research team referencing the American Academy of Pediatric Dentistry guidelines and pre-tested for clarity. Baseline data encompassing socioeconomic status, demographic details, household composition, pregnancy-related particulars, dietary diversity and household expenses will be obtained from the HAPIN research electronic data capture server. Information related to the child such as age, gender, type of delivery, only child and feeding patterns will also be abstracted.

### Biomonitoring

Biomarker analyses are central component of this study, drawing on existing biomarker research associated with ETS exposure and health. Biomarkers related to tobacco smoke exposure, inflammation, oxidative stress and epigenetic changes have been chosen to inform mechanistic pathways to link ETS exposure and ECC occurrence and also predict future dental caries risk (figure 2). Standard protocols established for resource-limited field settings will be followed to ensure sample integrity when collecting biospecimens from children.[31]

### Biological matrix and biomarker selection

We will collect non-invasive or minimally invasive biospecimens including saliva, buccal cells, DBS and urine from a subset (n=120) of the HAPIN children.

### Saliva

Saliva is increasingly used as an alternative matrix for biomonitoring, particularly in children. Its collection is non-invasive, economical, easily accessible and does not necessitate medical personnel in field settings with limited resources. Saliva provides recent physiological information about an individual through organic and inorganic components from the blood. Studies have shown a high correlation (r=0.94–0.99) in COT levels between saliva and plasma in both smokers and non-smokers.[32 33] Further, saliva is the preferred choice for assessing cariogenic bacteria counts and antioxidant status related to ECC. Saliva will be collected by passive

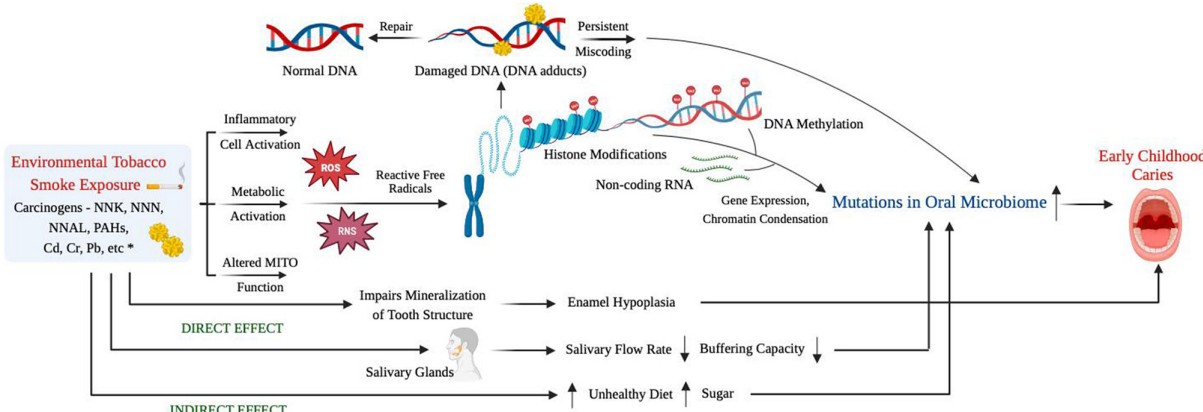

**Figure 2** Overall accepted mechanism linking environmental tobacco smoke exposure and early childhood caries occurrence. NNK, 4-(methylnitrosamino)-1-(3-pyridyl)-1-butanone; NNN, N′-nitrosonornicotine; NNAL, 4-(methylnitrosamino)-1-(3-pyridyl)-1-butanol; PAHs, polycyclic aromatic hydrocarbons; Cd, cadmium; Cr, chromium; Pb, lead; MITO, mitochondrial; ROS, reactive oxygen species; RNS, reactive nitrogen species

drool from children ensuring no food or drink has been consumed one hour into the visit. They will be requested to sit still and silent in a comfortable position for 2–3 min. Saliva obtained in the first 10 s will be discarded while that gathered thereafter within 5 min (approximately 1–2 mL) will be pooled in a single collection vial. We will also collect saliva samples from mothers for correlation with COT, 3HC levels, and bacterial count assessments.

## Buccal cells

Oral cell DNA is readily obtainable and constitutes a physiological domain impacted by ETS. ORAcollect for paediatrics (OC-175, DNA Genotek, Ontario, Canada) DNA collection kit will be used to obtain buccal cells. The sponge tip will be positioned comfortably in the mouth and rubbed along both lower gums in a back-and-forth motion. After collection, the sponge tip will be placed into its corresponding tube containing the Oragene solution and gently mixed. Genomic DNA will be extracted with the prepIT•L2P kits from DNA Genotek and sample aliquots will be stored at −80°C until analyses.

## Urine

Urine is a matrix of choice for environmental chemical metabolites that inform about biotransformation and internal exposure. Convenience or spot urine samples will be collected in sterile cups following ECC assessment. Urine will be aliquoted into cryochill vials and stored in deep freezers at −80°C until further analyses. A small aliquot will be used to measure creatinine using a semi-automatic biochemistry analyzer (Chem 7, Erba Mannheim, London, UK) through spectrophotometry.

In addition, specific gravity will be measured with a urine refractometer PAL-10S (Atago Co, Tokyo, Japan).

## Dried blood spots

We will collect routine blood samples as DBS on a five-spot Guthrie filter paper card (Whatman #903). This approach addresses the challenges associated with collecting, transporting and storing venipuncture samples.[34] Also, DBS blood collection in children is minimally invasive, cost-effective and requires low blood volume. In brief, the field staff will collect DBS samples from the children via finger prick after hand warming and arm massage to stimulate sufficient capillary blood flow for collection. The puncture site will be disinfected, and a sterile lancet will be used to puncture the ring or middle finger of the child's non-dominant hand. The first blood drop will be wiped away and subsequent blood drops will be applied to filter paper. Following sample collection, a sterile gauze pad will be pressed against the site until the bleeding stops. Samples will be air-dried for 24–48 h and then placed in a biospecimen bag with desiccant and a humidity indicator card.[34 35]

### Biomarker measurements

The choice of biomarker under consideration in each matrix, the analytic technique, and its significance to the objectives of the CARES study are summarised in table 1. The analytic protocol of each biomarker is provided in the sections below.

### Biomarkers in saliva

To evaluate saliva flow rate, the collected saliva will be quantified in millilitres per minute, and pH levels checked

**Table 1** List of biomarkers and summary of analysis scheme planned in the CARES–HAPIN study

| Biomarker | Matrix | Subject | Informs about | Method | Reference |
|---|---|---|---|---|---|
| Cotinine 3-hydroxycotinine | Saliva and urine | Child/mother | Recent exposure to ETS | LC-ESI-MS/MS | Sharma et al[37]; McGuffey et al[45] |
| Metals | Saliva and DBS | Child | Exposure to ETS | QQQ-ICP-MS | Gatzke-Kopp et al[38]; Langer et al[63] |
| Malondialdehyde | Saliva | Child/mother | Oxidative damage to membranes | Immunoassay | ELISA kit |
| Uric acid | Saliva | Child/mother | Cellular metabolism | Immunoassay | ELISA kit |
| Total antioxidant capacity | Saliva | Child/mother | Antioxidant capacity | FRAP | ELISA kit |
| Reduced glutathione | Saliva | Child/mother | Antioxidant capacity | Immunoassay | ELISA kit |
| *Streptococcus mutans, Lactobacilli* | Saliva | Child/mother | Cariogenic bacterial load | TaqMan-PCR | Yoshida et al[36] |
| DNA methylation | Buccal cells | Child | Epigenetic modification | BeadArray | Triche et al[40] |
| DNA adduct | Buccal cells | Child | Exposure to TSNAs and PAHs | LC-ESI-MS/MS | Stepanov et al[41]; Chuang et al[42] |
| 8-hydroxy-2'-deoxyguanosine | Urine | Child | Oxidative stress | LC-ESI-MS/MS | McGuffey et al[45] |
| PAH metabolites | Urine | Child | Exposure to PAHs | HPLC-FLD | Puttaswamy et al[53] |
| 4-(methylnitrosamino)-1-(3-pyridyl)-1-butanol, N′-nitrosonornicotine | Urine | Child | Exposure to ETS | LC-ESI-MS/MS | Nikam et al[56] |
| 25 hydroxyvitamin D2 and 25 hydroxyvitamin D3 | DBS | Child | Vitamin D reserve | LC-ESI-MS/MS | Eyles et al[59] |
| Telomere length | DBS | Child | Cellular ageing | qPCR assay | ScienCell, California, USA |

CARES–HAPIN, Caries Risk from exposure to Environmental tobacco Smoke–Household Air Pollution Intervention Network; DBS, dried blood spots; ELISA, enzyme-linked immunosorbent assay; ETS, environmental tobacco smoke; FRAP, ferric reducing antioxidant power; HPLC-FLD, high-performance liquid chromatography and fluorescence; LC-ESI-MS/MS, liquid chromatography-electrospray ionisation-tandem mass spectrometry; PAHs, polycyclic aromatic hydrocarbons; qPCR, quantitative-polymerase chain reaction; QQQ-ICP-MS, triple quadrupole inductively coupled plasma mass spectrometer; TSNAs, tobacco-specific nitrosamines.

within 30 min of collection. Salivary counts of *Streptococcus mutans* and *Lactobacilli* will be determined using TaqMan PCR assay.[36] Total antioxidant capacity and reduced glutathione levels will be measured using the ferric reducing antioxidant power assay and immunoassay technique, respectively.

*Cotinine and 3-hydroxycotinine:* The half-life of COT (~16 h) is longer compared to nicotine (~2 h) and it is a predominant and stable metabolite of nicotine in body fluids. A simple, rapid and sensitive method will be used to measure COT and 3HC simultaneously by the liquid chromatography-electrospray ionisation-tandem mass spectrometry (LC-ESI-MS/MS) technique. An aliquot of saliva will be centrifuged in a cooling microcentrifuge (Eppendorf 5430R) at 5000 rpm for 10 mins. An equal volume of acetone will be added to the supernatant, with the final volume adjusted to 1 mL using 0.1% formic acid in methanol solution. The resulting mixture will be vortex mixed and passed through Millex-HV syringe filters (4 mm, 0.45 μm) prior to analysis by LC-ESI-MS/MS. The limit of detection (LOD) for this method is 0.5 ng/mL, with a recovery of 99.6% and relative standard deviation (RSD) ranging from 9.2% to 9.5% for salivary COT in smoker's and non-smoker's samples.[37]

*Metal(loid)s analysis:* We will analyse trace metals in saliva and investigate their correlation with salivary COT to confirm if exposure to ETS contributes to the presence of metals in children. A recent study with 6 to 7-year-old children from the Family Life Project revealed positive correlations between salivary COT and lead, zinc and copper, providing evidence for ETS as a source of trace metals.[38] Briefly, 0.5 mL of saliva will be weighted and digested overnight in 2 mL of ultrapure concentrated $HNO_3$. Subsequently, samples will be centrifuged at 15 000 rpm for 5 min at 4°C. The supernatant will be diluted at a 1:100 ratio with 1% $HNO_3$ with indium as the internal standard, and metal concentrations will then be measured on a single run of triple quadrupole inductively coupled plasma mass spectrometer (QQQ-ICP-MS, Agilent 8900). Quality control (QC) measures will be employed to ensure accurate concentration estimates: procedural blanks, replicates, initial and continuous calibration verification. To assess the validity and reliability of measuring metals in the saliva of children, we will conduct a pilot analysis of paired samples (n=10) of saliva, venous blood and DBS collected from healthy adult volunteers.

*Malondialdehyde (MDA) and uric acid (UA):* Measurement of MDA and UA is essential for studying mitochondrial functions as these components play a key role in energy metabolism and production of reactive oxygen species. Saliva samples will be centrifuged at 5000 rpm for 10 min at 4°C and the supernatant will be used directly to measure MDA by colorimetric assay kits following the manufacturer's procedure. UA will be quantified by Salimetrics UA assay kit (#1-3802) which has a wide assay range from 0.07 to 20 mg/dL and a high sensitivity of 0.07 mg/dL LOD.

## Biomarkers in buccal cells

*DNA methylation (DNAm):* Buccal cell DNA will be used to analyse DNAm with the Infinium Methylation EPIC V.2.0 BeadChip which targets over 935 000 CpG sites. To ensure QC, replicates and randomisation of samples across chips and plates will be employed. DNAm data will be processed using the minfi R package.[39] Samples not meeting certain criteria (eg, duplicate samples, poor individual call rate <0.98 or genotyping/sex mismatch) will be excluded. Background correction, dye bias adjustment through the 'noob' method and quantile normalisation will be performed according to established procedures.[40]

*DNA adducts:* We will focus on quantifying DNA adducts formed by TSNAs and PAHs using the LC-ESI-MS/MS method as described elsewhere.[41 42] TSNAs such as 4-(N-nitrosomethylamino)−1-(3-pyridyl)−1-butanone (NNK) and N′-nitrosonornicotine (NNN) are found together in environments contaminated by tobacco smoke. The metabolic process involving α-hydroxylation at the 2'-position of NNN or the methyl group of NNK leads to the formation of a reactive intermediate (pyridyloxobutylate) that binds with DNA, resulting in pyridyloxobutyl (POB) DNA adducts. Certain POB-DNA adducts can break down under strong acidic conditions, releasing 4-hydroxy-1-(3-pyridyl)−1-butanone. The primary DNA adduct formed by PAH is benzo(a)pyrene diol epoxide. Researchers have found detectable levels of DNA adducts derived from PAHs and TSNA in the oral cells of smokers and non-smokers exposed to ETS.[43 44]

## Biomarkers in urine

*Cotinine and 3-hydroxycotinine:* Urine (200 μL) spiked with labelled internal standards will be enzymatically hydrolysed at 37°C overnight in an orbital shaker to release the conjugated analytes. We will add 0.85 mL of cold acetone to each sample and refrigerate for over 10 min, followed by centrifugation at 4000 rpm for 2 min to eliminate precipitates of salts, protein and exogenous enzymes. The top urine/acetone layer will then be transferred into a new vial and partially dried to remove the acetone. A 10 μL of the remaining supernatant urine will be directly injected into the LC-ECI-MS/MS system for simultaneous determination of total COT and 3HC.[45] Calibrants, solvent blank, matrix blank, and duplicate QC samples will be concurrently analysed.

*Marker of oxidative stress:* ETS constituents undergo metabolic activation, forming diol epoxides. These reactive compounds generate reactive oxygen species that can interact with DNA base pairs resulting in DNA damage.[46] Over 30 different types of base modifications have been identified, of which 8-hydroxy-2'-deoxyguanosine (8OHdG) is widely recognised as a biomarker of 'oxidatively damaged DNA'.[47] Researchers have measured levels of 8OHdG in population-based studies using different matrices like venous blood, urine, placenta, cord blood and saliva to investigate the potential health effects caused by ETS exposure.[48–51] 8OHdG will be measured using the LC-ESI-MS/MS method.[45]

*PAH metabolites:* PAHs, known for their carcinogenic and mutagenic properties, are mainly formed as a result of incomplete combustion of tobacco and other organic materials.[52] PAH metabolites including 2-naphthol, 1-hydroxypyrene and other relevant metabolites of 4-ring or less will be determined by high-performance liquid chromatography and fluorescence detection previously published and cross-validated by LC-ESI-MS/MS technique.[53] The method LOD is <0.05 ng/mL and RSD is <5%. Creatinine will be measured and used for normalisation of urine dilution.

*4-(methylnitrosamino)–1-(3-pyridyl)–1-butanol (NNAL):* Children exposed to ETS inhale TSNAs which are potent organ-specific carcinogens. The primary metabolite NNAL serves as an important biomarker of both exposure and effect and has been reported in pregnant women and children exposed to ETS.[54 55] We will employ a previously developed method for simultaneous determination of total NNAL and NNN by LC-ESI-MS/MS. This method has reported LOD, accuracy and precision of 0.05 pmol/mL, 110.9% and 2.9%, respectively, for total NNAL.[56] These will be some of the first estimates of TSNA metabolites in children exposed to ETS from India.

### Biomarkers in DBS

*Haemoglobin (Hb) and vitamin D:* Iron deficiency in children increases the risk of ECC.[57] We plan to conduct Hb measurements using point-of-care devices (HemoCue Hb 201 system) for immediate results that can be shared with physicians if necessary.

Vitamin D plays a crucial role in oral health. Inadequate vitamin D levels during periods of tooth development affect tooth calcification, predisposing enamel hypoplasia and ECC.[58] 25 hydroxyvitamin D2 and 25 hydroxyvitamin D3 will be extracted from 6 mm DBS, derivatised with 4-phenyl-1,2,4-triazoline-3,5-dione before analysis by LC-ESI-MS/MS method.[59]

*Telomere length (TL):* Telomeres are terminal DNA-protein structures that maintain the stability and integrity of chromosomes.[60] They are at risk of damage from reactive oxygen species produced by ETS components. Studies have demonstrated a dose–response relationship between prenatal ETS exposure and TL attrition.[61 62] Briefly, genomic DNA extraction will be performed using the QIAcube and QIAamp Mini Kit according to the manufacturer's protocol. The average relative TL represented by TL/single copy gene (T/S) ratio will be determined by quantitative-polymerase chain reaction using the absolute human TL quantification assay kit (ScienCell, California, USA) in conformance with minimum reporting recommendations of the Telomere Research Network, Tulane University School of Medicine, Louisiana, USA. Each sample will be run in duplicate, with a standard curve and two internal QC samples included on each plate.

*Metal(loid)s analysis:* A full spot which corresponds to approximately 60 µL of capillary blood will be excised with a ceramic blade into Teflon vials. A blank spot will be cut from the same filter card to correct for background contamination. The weights of the digestion vials before and after will be documented to consider the precise mass of digested blood. Metal concentrations will be measured using QQQ-ICP-MS on a single run. The certified reference standard, Seronorm Serum L2 (Sero, Norway) will be analysed once daily as a QC sample. Multiple isotopes of each metal will be monitored along with blanks, duplicates and standard reference material to ensure method accuracy and precision.[63]

### Fluoride concentration in water

Fluoride deficiency, especially widespread in socioeconomically disadvantaged areas, is a well-established risk factor for ECC. Therefore, we will gather information on toothpaste usage and collect water samples for fluoride analysis. Water samples will be collected from a subset of 50 homes and 25 common drinking water sources in the community. The concentration of fluoride in the water will be measured using a fluoride ion-selective electrode (Orion 9609BNWP, Thermo Scientific, USA) connected to a bench-top analyser. The data obtained will be recorded in ppm. Prior to each measurement, the electrode will be calibrated with four standard fluoride solutions: 1, 10, 50 and 100 ppm.

### Data management and reporting

To maintain confidentiality, patients will be allocated unique identification numbers which will be used on all data collection forms and questionnaires. Data obtained will be secured/encrypted, computer data files will be password protected, and other study materials will be stored in locked file cabinets. We will collate the data sheets and create a final dataset in Excel. This will undergo thorough validation to ensure coherence and identify any potential errors.

### Quality assurance/quality control

Each filter card and collection cup will be labelled with the unique participant IDs and bar codes. To ensure a continuous cold chain, temperature-controlled Credo ProMed portable cooler bags will be used for collecting and transporting samples from field sites to the laboratory. On their arrival at the HAPIN BioMonitoring laboratory at SRIHER, the samples will undergo a cross-check against their tagged IDs and custody forms to verify accuracy before storing in deep freezers for long-term bio-archival. Trained personnel will record the quantity, quality and condition of the DBS. Any smudged or double-dropped spots will be documented. If a spot is too small to accommodate a single small hole punch (3 mm), it will be deemed inadequate. Conversely, spots that can accommodate a single large hole punch (6 mm) will be considered suitable and those that fill the circle will be categorised as excellent.[64] Three to four aliquots of appropriate volumes will be prepared for saliva and urine to minimise multiple freeze-thaw cycles and facilitate long-term storage in a −80°C deep freezer. Field blanks and

transport blanks will be employed to detect any potential contamination during sample collection, handling and transport processes.

## STATISTICAL ANALYSIS

Baseline characteristics will be summarised using frequency and percentage for categorical variables and mean and SD for continuous variables with symmetric shape, otherwise median and IQR. The external validity in mean differences between children with and without ECC will be assessed using t-tests and for the difference in proportions $\chi^2$ tests of homogeneity will be used. To group similarly affected tooth surfaces, Ward's minimum variance method, a type of agglomerative hierarchical clustering will be applied. Each surface will start as its own cluster, and similar clusters will then be successively merged to form a clustering hierarchy. The similarity among clusters will be evaluated using the Euclidean distance.[65] To analyse the association between the timing of ETS exposure (in utero, infant and current) and stages of ECC, generalised linear mixed models with a logit link function will be employed. We will fit adjusted models for each outcome, accounting for potential confounders (ie, maternal age, parental educational levels, household income, child age, gender, preterm birth/low birth weight and yearly household income). To examine whether the prevalence of ECC increases with pack months of ETS exposure at home, a logistic regression model will be used. Quantile g-computation will be applied to estimate the association of metal mixtures with dental caries using the R package qgcomp.[66] We will explore the use of advanced statistical methods to assess interrelationships between ETS biomarkers, oxidative stress and ECC outcome. We will also implement novel statistical methodology and machine learning algorithms to build epigenetic predictors of ETS exposure and ECC severity. Furthermore, we will use bioinformatic tools to map sites of epigenetic modifications between cases and controls. All statistical analyses will be performed using R V.4.2.0[67] and bioinformatic packages will be downloaded from Bioconductor.[68]

### Study power

Sample size calculation was based on a previous study which reported prevalence of ECC in the Indian population as 49.6%.[18] We estimated a sample size of 384 participants with an absolute precision of 5% and a 95% CI. The CARES–HAPIN cohort consisting of 735 children has adequate statistical power to detect any association between ETS exposure and ECC.

### Patient and public involvement

Lay summaries of the results will be provided to patients, caregivers and the wider community after the final report is prepared. Educational infographics illustrating the adverse effects of ETS exposure on pregnant women and fetal development will also be created for distribution

at primary health centres. The quantitative exposure–response relationship derived from the study will have valuable implications for both scientific inquiry and policy formulation. Therefore, we will aim to make this data widely accessible while upholding participants' consent and confidentiality agreements.

## ETHICS AND DISSEMINATION

CARES study protocol has been approved by the institutional review board of the participating institute (IEC-NI22/JUL/83/82). Agreements for sharing materials and data will be formulated based on the preferred standards of the institution and HAPIN guidelines. The conduct of this study will align with ethical principles outlined in the Belmont Report and follow Good Clinical Practice guidelines in line with the Indian Council of Medical Research.

Written informed consent will be obtained from the parent(s)/legal guardian(s) of all children. Consent protocol entails providing participants with comprehensive information regarding the aims, methods, potential risks and benefits of participation. Parents will be informed that their child's participation is voluntary, and even if they choose not to, their child will still receive the necessary care in accordance with HAPIN guidelines. Moreover, parents will be reassured that appropriate measures are in place to safeguard their child's personal information and that any biospecimens collected will only be used for the specific study unless otherwise consented by them. They will be made aware that in the event an abnormality is detected during the analyses of samples, reports will be communicated to them, and further testing and medical interventions be advised as necessary.

The study findings will be published in peer-reviewed journals and disseminated through a national-level symposium involving researchers, subject experts, and healthcare providers.

### Practical lessons from CARES–HAPIN pilot study

A preliminary investigation was conducted with 21 children to evaluate the adequacy of the research procedures (i.e., clinical assessment, questionnaire administration and biospecimen collection). Half of the children exhibited high levels of dental caries, with an average of 37.1% affected surfaces. This pilot study enabled us to establish procedures for analysing COT, 3HC and 8OHdG in urine as well as DNA extraction, purification and sequencing in DBS. Moreover, it was observed that a majority of children and parents were willing to participate in this study.

We encountered several challenges throughout the process, which provided important lessons:

► It is crucial to ensure that the study instruments are not only translated into relevant local language but also culturally appropriate for the population being studied. Understanding and addressing cultural barriers in research hold significance.

► In longitudinal cohort studies, participant tracking can be challenging, especially with migrant populations.

Maintaining regular communication with participants and obtaining contact details of individuals who are familiar with their whereabouts can prove beneficial. In our study, we found that obtaining the contact details of the child's mother was notably advantageous.

► Research staff must be adaptable and willing to work beyond the regular hours. Also, evaluations at primary healthcare centres may not always be suitable, necessitating the establishment of alternatives such as anganwadis (pre-schools) and/or home visits.

► Special consideration should be given to biological specimen collection in remote or rural environments, ensuring strict adherence to standard operating procedures for sample integrity. Careful handling and storage of biological samples are paramount to gather extensive information. Regular quality assurance and QC measures should also be implemented.

## DISCUSSION

ECC is an insidious adversary, affecting millions of children worldwide. If the study's hypothesis is supported, it will help to elucidate the potential biological pathways through which ETS exposure impacts dental caries, impelling the development of more targeted ECC interventions and therapies. Additionally, to our knowledge, this will be one of the first studies to demonstrate how ETS exposure can influence epigenetic programming with potential downstream consequences for developing dental caries later in life.

The strength of our approach includes leveraging the existing high-quality data and resources from the HAPIN cohort, which will enable us to cost-effectively recapitulate information on the prenatal and early postnatal environmental milieu. It has an established comprehensive survey data on ETS exposure during pregnancy (maternal report only) and childhood (child and maternal report); a wealth of covariate data to rigorously adjust for confounders and precision variables across analyses; and a field network system to undertake robust ECC assessment. The HAPIN cohort has also created a databank with bioassay results (urinary COT, 3HC, 8OHdG and PAH metabolites) and a biorepository, thereby, providing a resource to investigate modifiable stressors that impact the child's trajectory of dental caries. Furthermore, it is sufficiently powered to analyse outcomes related to harmful environmental exposures. This serves as a groundwork for an in-depth analysis of the critical window or 'timing' of ETS exposure in relation to ECC occurrence. The research team is composed of experts who have carried out seminal work in all the relevant research domains on which the study is based, including environmental health and epidemiology, human biomonitoring, epigenetics and biostatistics. This ensures that the study maintains both high standards in design and execution, while also remaining practical and pragmatic throughout. We recognise that

the successful implementation of the study hinges on a meticulous collection, storage and analyses of biospecimens. The experience gained from conducting the CARES– HAPIN pilot study has proved to be valuable in refining the main research endeavour. As a result, minor adjustments have been made to improve how information is presented to parents and children, as well as streamline the biospecimen collection. Our work will provide insight into how simultaneous environmental exposures at critical time points interact among themselves, with individual characteristics (e.g., genetics) and epigenetics—a key component to reduce the ECC prevalence. This will highlight opportunities for policy recommendations.

**Author affiliations**
¹Department of Environmental Health Engineering, Faculty of Public Health, Sri Ramachandra Institute of Higher Education and Research (Deemed to be University), Chennai, Tamil Nadu, India
²Department of Pediatric and Preventive Dentistry, Dr. D.Y. Patil Dental College and Hospital, Dr. D.Y. Patil Vidyapeeth, Pune, Maharashtra, India
³Department of Epidemiology and Population Health, Stanford University, Stanford, California, USA
⁴Gangarosa Department of Environmental Health, Rollins School of Public Health, Emory University, Atlanta, Georgia, USA
⁵Department of Biostatistics, St John's Medical College, Bengaluru, Karnataka, India

**Contributors** NP, SSP and AC conceptualised the project and were involved in developing study methodology and funding acquisition. SSP and NP drafted the manuscript. AC, DBB, SG and KB edited the manuscript contributing to critical content expertise.

**Funding** The CARES study is supported by the Fogarty International Center (D43TW010540) with participation from the National Institute of Dental and Craniofacial Research. The HAPIN trial is funded by the US National Institutes of Health (cooperative agreement 1UM1HL134590) in collaboration with the Bill & Melinda Gates Foundation (OPP1131279).

**Competing interests** None declared.

**Patient and public involvement** Patients and/or the public were involved in the design, or conduct, or reporting, or dissemination plans of this research. Refer to the Methods section for further details.

**Patient consent for publication** Not applicable.

**Provenance and peer review** Not commissioned; externally peer reviewed.

**ORCID iDs**
Sneha S Patil http://orcid.org/0000-0001-6698-2711
Naveen Puttaswamy http://orcid.org/0000-0002-8221-1682
Dana Boyd Barr http://orcid.org/0000-0002-5566-2138

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
