## [Reviewer comments · BMJ Open]

ARTICLE DETAILS

TITLE (PROVISIONAL)	Protocol for CARES - HAPIN: An ambidirectional cohort study on exposure to environmental tobacco smoke and risk of early childhood caries
AUTHORS	Patil, Sneha; Puttaswamy, Naveen; Cardenas, Andres; Barr, Dana; Ghosh, Santu; Balakrishnan, Kalpana

VERSION 1 – REVIEW

REVIEWER	Oyapero, Afolabi Lagos State University College of Medicine, Preventive Dentistry
REVIEW RETURNED	24-Feb-2024

GENERAL COMMENTS	The manuscript appears well written. The writing style, grammar, tenses and punctuation are adequate. Each section of the manuscript is clearly presented. The introduction has a clear background, appropriate review of literature and it clearly identifies the gap in knowledge that justifies the study. The methodology is also detailed and the study instruments and analytic method proposed is adequate. The researchers should however specify if the consent obtained clearly stated that genetic analysis will be done and that the collected samples will not be utilized for any other study without the consent of the participants.
--

REVIEWER	Martins-junior, Paulo Federal University of Minas Gerais
REVIEW RETURNED	12-Mar-2024

GENERAL COMMENTS	This cohort study protocol will evaluate the association between environmental tobacco smoke and early childhood caries. The idea is very original and will add substantial knowledge to scientific community. In general, the protocol is very well written and very well designed. I have only few suggestions to authors: I would like to suggest that authors include the assessment of pufa index to evaluate clinical consequences of untreated ECC as well as a oral health-related quality of life assessment by ECOHIS.
--

VERSION 1 – AUTHOR RESPONSE

Reviewer 1

1. Specify if the consent obtained clearly stated that genetic analysis will be done and that the collected samples will not be utilized for any other study without the consent of the participants.

Response: The following statement has been added - 'Moreover, parents will be reassured that appropriate measures are in place to safeguard their children's personal information and that any biospecimens collected will only be used for the specific study unless otherwise consented by them.' The same has now been mentioned in the revised manuscript.

Reviewer 2

1. Include the assessment of PUFA index to evaluate clinical consequences of untreated ECC as well as a oral health-related quality of life assessment by ECOHIS.

Response: As recommended by the reviewer, we have revised the methodology section to include details on PUFA index and oral health related quality of life.

The following statements have been added.

- The PUFA index will be employed to evaluate the consequences of untreated dental caries,³⁰ while the Plaque Index developed by Silness & Loe (1964) will be utilized to assess the plaque level. (30. Monse B, Heinrich-Weltzien R, Benzian H, Holmgren C, van Palenstein Helderma W. PUFA -an index of clinical consequences of untreated dental caries. Community Dent Oral Epidemiol. 2010;38(1):77-82)
- Trained bilingual staff members will interview parents to collect information via detailed questionnaires on oral hygiene (initiation of brushing, frequency, parental supervision, use of fluoride toothpaste usage), diet (frequency of sweets consumption, eating before sleep), dental attendance (dental visit frequency and reasons), receipt of medications and child's oral health-related quality of life using the Early Childhood Oral Health Impact Scale (ECOHIS). The questionnaire has been designed by the research team referencing the American Academy of Pediatric Dentistry guidelines and pre-tested for clarity.